# Beyond the ORF: Paralog-specific regulation of *RPS7/eS7* mRNAs via 3'-UTRs and promoter sequences

**Sachiko Hayashi** *, **Tohru Yoshihisa**

Graduate School of Science, University of Hyogo, Ako-gun, Japan

* shayashi@sci.u-hyogo.ac.jp

## Abstract

In a classical view, each paralogous ribosomal protein (RP) is equally synthesized and integrated into the ribosome. Therefore, RP-paralog mRNAs are generally believed to be similarly regulated on their transcription and/or stability. In this paper, we report that two Rps7p/eS7 paralogs of *Saccharomyces cerevisiae* are differently regulated; deletion of *RPS7A* upregulates *RPS7B* paralogous mRNA expression but not *vice versa*. Their 3'-UTR sequences critically regulated the stabilities of both *RPS7A* and *RPS7B* mRNAs. Alterations in these sequences led to a diminished expression of *RPS7A* and *RPS7B* mRNAs in a transcript-dependent manner, suggesting that *RPS7*-paralog mRNAs have different properties for their expression and/or stability. The C-terminal tagging of the ORF and mutation analyses in the 3'-UTR indicate that both *RPS7*-paralog mRNAs critically rely on their 3'-UTRs for mRNA expressions and/or stabilities. We also found that activities of both *RPS7A* and *RPS7B* promoters are regulated by abundance of Rps7Ap and that Fhl1p, a key transcriptional regulator of RP genes, is essential for transcription of *RPS7B* but not *RPS7A* while simultaneous deletion of a consensus sequence for Fhl1p in the *RPS7A* promoter region and the *FHL1* gene completely abolishes the promoter activity. These results indicate that yeast has a distinct buffering system for Rps7p production between the two *RPS7*-paralogs, which is sensitive to variation on their 3'-UTRs and is partially mediated in a transcription-dependent manner.

## Introduction

Ribosomes are central molecular machines dedicated to translation in all of life. In *Saccharomyces cerevisiae*, 79 ribosomal proteins (RPs) and four ribosomal RNA (rRNA) are assembled into the ribosome with the assistance of 76 different small nucleolar RNAs and more than 200 assembly factors [1,2]. Fifty nine of the 79 RPs are encoded by a pair of paralogous genes named A and B, and these RP paralogs are considered to be generated mainly by whole genome duplication [3]. Indeed,

**Data availability statement:** All relevant data are within the manuscript and its Supporting Information files.

**Funding:** This work was supported by JSPS KAKENHI (https://www.jsps.go.jp/english/) Grant number JP20K06491 (to S. H), 17H05672 and 23K18100 (to T. Y.), Takeda Science Foundation (https://www.takeda-sci.or.jp/en/), Hyogo Science and Technology Association (https://hyogosta.jp/), Japan (to S. H.; 6082), and the Institute for Fermentation, Osaka (https://www.ifo.or.jp/; to S. H.; G-2025-2-032). The funders had no role in study design, data collection and analysis, decision to publish, or preparation of the manuscript. There was no additional external funding received for this study.

**Competing interests:** The authors have declared that no competing interests exist.

**Abbreviations:** RP, ribosomal protein; NMD, nonsense-mediated decay; UTR, untranslated region.

21 of the 59 RP-paralog pairs encode identical proteins while the others show high sequence similarities [4]. These paralog pairs are thought to serve as a repository for buffering deleterious changes in either copy [5], *e.g.,* to permit adjustment of the RP dose to match that of rRNA synthesis, thereby ensuring ribosome assembly. Emerging evidence showed that RP-paralog pairs possess distinct or even opposite expression patterns and paralog-specific functions [6]. Thus, heterogeneous RP-paralog pairs not only maintain a proper ribosomal level by complementing each other's expression as dose amplification but also offer the potential to generate heterogeneous ribosomes by swapping distinct RP isoforms to fine-tune the translational program known as paralog-specific regulation [6,7]. Further, deletion of RP genes can lead to aneuploidy in yeast as a compensatory response to the loss of the RP copies [4,8]. Cells prefer to duplicate chromosomes containing the remaining RP copies to restore balance, enhancing their fitness despite the aneuploidy.

To date, yeast RP paralog-specific differences were observed on a wide range of cellular and metabolic functions: sensitivity/resistance against various types of drugs, localization and translation of *ASH1* mRNA [9] as well as translational accuracy of the luciferase-based reporter mRNAs [10]. Quantitative mass spectrometry analysis revealed a compositional change of a Rpl8p paralog pair in the 80S ribosome in response to a shift from glucose to glycerol media [11]. *RPL7* paralogous genes generate two types of acetylated RPs, Rpl7Ap and Rpl7Bp, leading to modulated mRNA translation depending on the ORF length and achieving different drug resistance [12]. Although reducing the availability of duplicated RP genes does not impair overall ribosome function or viability, it still can lead to distinct phenotypes [4,13,14]. In addition, these duplicated RPs are differently expressed, and such differential expression under normal conditions is mainly achieved not only by transcriptional regulation but also by post-transcriptional events, such as splicing, mRNA stabilization and translation control [3,7]. To comprehensively understand the intricate regulatory mechanisms governing these paralog-specific disparities in yeast RP genes and their impacts on cellular processes and phenotypes, further investigations and detailed analyses are imperative.

Here, we revealed paralog-specific regulations of *RPS7* mRNAs, encoding a eukaryotic ribosomal protein Rps7p/eS7. Originally, the tRNA-intronless *tl(caa)Δint** mutant (see Results in detail) that found to have a long insertion in the *RPS7B* 3'-UTR that destabilized *RPS7B* mRNA by nonsense-mediated decay (NMD). This mutation, akin to the *rps7bΔ*, did not induce *RPS7A* expression, while the *rps7aΔ* mutation upregulated *RPS7B* mRNA to raise Rps7Bp production. Modified 3'-UTRs in both cases of *RPS7*-paralog mRNAs impaired the corresponding mRNA level in a transcript-dependent manner. Additionally, reporter gene assay using the promoter regions of *RPS7A* and *RPS7B* revealed different contributions of Fhl1p, a prominent transcription factors almost exclusively associated with ribosomal protein genes in yeast, to their promoter activities. This suggests that transcription also forms an overlooked layer for paralog-specific regulation. Overall, yeast has a different buffering system for Rps7p production between the two *RPS7*-paralog mRNAs that are sensitive to a variation on their 3'-UTRs.

## Materials and methods

### Strains, primers, plasmids, and media

*S. cerevisiae* BY418-based strains are summarized in S1 Table in S1 File. Primers and plasmids used in this study are listed in S2 and S3 Tables in S1 File, respectively. The plasmids utilized in the reporter gene assay were created using the pSA144 plasmid (gift from Prof. Toshifumi Inada, The University of Tokyo, Japan) as the backbone. Details of plasmid construction are also described in S3 Table in S1 File. Standard yeast genetic techniques and other molecular biological techniques used here are essentially described in [15,16]. Gene disruptions and marker insertions were carried out using a PCR-based strategy; a marker cassette with appropriate flanking sequences of a target gene was amplified by PCR, and the target gene on the chromosome was replaced with the PCR fragment by homologous recombination. A C-terminally HA-tagged *RPS7A* fragment was similarly made using PCR with a 3 × HA tagging cassette containing an *ADH1* terminator and a *CgHIS3* marker, and were integrated into the indicated parental strains, leading to *RPS7A-HA::CgHIS3* strains. The *RPS7B-FLAG* strains were generated by the two-step gene replacement strategy; the chromosomal *RPS7B* gene was first disrupted by a *URA3* marker, and subsequently the disrupted *RPS7B* gene region was replaced with an amplified *RPS7B-3×FLAG* fragment. Final transformants expecting *ura3 RPS7B-3×FLAG* strains were selected by 5-fluoroorotic acid plates, and correct integration was confirmed by sequencing. For introducing the mutated *RPS7B* 3'-UTR appearing in TYSC2148 (*rps7b-102* mutation) into a *tL(CAA)N* wild-type strain, the corresponding DNA fragment with the *RPS7B* second exon was amplified from TYSC2148 chromosomal DNA using RPS7B_int_242_fw and eS7B_3UTR_158–227 primers, and then was integrated into SHSC0322 (*tL(CAA)N rps7bΔexon::URA3*) to replace the corresponding region including the *URA3* marker. Correct integration was confirmed by sequencing. For constructing SHSC0633 with *fhl1Δ::CgHIS3*, gene disruptions and marker insertions were conducted using the PCR-based strategy described above.

All strains were essentially grown at 30°C in YPD [1.0%(w/v) yeast extract, 2.0%(w/v) polypeptone, and 2.0%(w/v) D-glucose]. In case of solid media, final 2.0%(w/v) agar was added.

### Crude RNA preparation, northern blotting, and RT-PCR

RNA preparation and northern blotting were performed as essentially described in [16]. Total RNAs from logarithmically growing yeasts were extracted by the hot phenol method with the Na-acetate/SDS buffer [50 mM Na-acetate, pH 5.2, 10 mM EDTA, 1.0%(w/v) SDS] and Phenol, Saturated with Citrate Buffer [pH 4.5] (Nacalai Tesque, Kyoto, Japan). RNA (5.0 μg) per lane was analyzed on 1.2% agarose gel with formaldehyde, and transferred to Hybond N$^+$ charged nylon membrane (GE Healthcare, Chicago, Illinois, USA) by capillary transfer. Antisense RNA probes of *RPS7A* and *RPS7B* were labeled with digoxigenin using DIG Northern Starter Kit (Roche Diagnostics, Basel, Switzerland). After hybridization in DIG Easy Hyb (Roche Diagnostics) at 68°C followed by immunodecoration with ALP-labeled anti-digoxigenin IgG, signals were visualized by CDP-star (Roche Diagnostics). Chemiluminescence was captured and analyzed by a cooled CCD camera system EZ-Capture and imaging analysis software CS Analyzer 3 (ATTO, Tokyo, Japan). For RT-PCR, poly-A RNAs in total yeast RNA samples prepared from appropriate strains were converted into cDNAs by reverse-transcription using SuperScript III reverse transcriptase (Thermo Fisher Scientific, Waltham, Massachusetts, USA) with oligo dT$_{20}$ as a primer. Using the cDNA, 3'-UTR regions of *RPS7B* alleles *etc.* were amplified by PCR with appropriate primer sets and Ex-Taq DNA polymerase (Takara Bio, Otsu, Japan). Primers used in RT-PCR are listed in Table S4 in S1 File (Supporting Information).

### Total protein extraction and western blotting

Crude proteins from mid log-phase yeast cells (0.5 OD$_{660}$ unit) were prepared by alkaline extraction [17], and separated with 10% SDS-polyacrylamide gel followed by transfer to a PVDF membrane (Immobilon P; Millipore, Burlington, Massachusetts, USA). The primary antibodies used here were anti-Srp1p rabbit polyclonal antibodies, anti-HA tag mouse

monoclonal antibody (TANA2: MBL, Tokyo, Japan), and anti-DYKDDDDK tag mouse monoclonal antibody (1E6: FUJIFILM Wako Pure Chemical, Osaka, Japan). After incubation with horseradish peroxidase-conjugated goat anti-rabbit or anti-mouse IgG, immunoblot signals were developed via home-made ECL system, and chemiluminescence signals were captured and analyzed as northern blotting.

## Results

### The *tl(caa)Δint\** strain operates NMD for *RPS7B* mRNA due to its aberrant longer 3'-UTR, whereas deletion of *RPS7A* results in the upregulation of *RPS7B*

We had previously generated the complete set of *S. cerevisiae* tRNA-intron deletion strains [18] using a pop-in-pop-out strategy with a *GALp-GIN11M86* marker [19]. The integration-marker cassette, after replacing an intron-containing tRNA gene on a chromosome with an intron-less one, was removed by homologous recombination between the *hisG* sequences, or between flanking sequences that were used for chromosomal integration of the tRNA gene using negative selection of the galactose-inducible toxic marker (Fig 1A). One of these tRNA intron-deletion strains, *tl(caa)Δint* (TYSC2148) that converted all the *tL(CAA)* genes in the parental genome to the intron-less ones, was found to have a 2.3-kb insertion of *loxP::CmR::hisG::loxP* located between the *tL(CAA)N/YNCN0007W* coding region and the *RPS7B/eS7B* ORF, while the reported genome-wide transcriptome data indicated that the marker was inserted into the 3'-UTR of *RPS7B* [20,21]. Hereafter, this *tl(caa)Δint* strain with the marker insertion is referred to as *tl(caa)Δint\**.

To examine effects of unintended marker insertion in the 3'-UTR of *RPS7B,* we performed northern analyses of *RPS7B* mRNA. The *RPS7B* band was clearly detected with *RPS7B*-specific probe in the wild-type lane (black triangle), whereas it was barely seen in the *tl(caa)Δint\** lane (Fig 1B, lanes 1 and 2). Total RNAs visualized by GelRed staining prior to northern analysis were similar between the two strains (Fig 1C). Thus, the insertion compromised expression of *RPS7B* mRNA selectively.

In certain eukaryotes, such as *S. cerevisiae*, mRNAs with aberrantly long 3'-UTRs, including artificially elongated variants, are main targets of NMD via the faux 3'-UTR mode [22,23]. To analyze whether this expression defect of *RPS7B* mRNA in the *tl(caa)Δint\** was caused by NMD, we deleted the *UPF1* gene, which encodes one of the core NMD factors [24,25]. As shown in Fig 1B longer transcript that was detected with the *RPS7B*-specific probe (white triangle) appeared in the *tl(caa)Δint\* upf1Δ* strain, but not in the *upf1Δ,* the wild type nor the *tl(caa)Δint\** (Fig 1B, middle and bottom panels, lanes 1–4). The marker insertion of the *tl(caa)Δint\** would elongate the 3'-UTR of *RPS7B* and render the *RPS7B* mRNA variant susceptible to NMD. We denoted to the *rps7B* mutant allele in the *tl(caa)Δint\** strain, characterized by the *loxP::CmR::hisG::loxP* insertion in the *RPS7B* 3'-UTR as *rps7b-102.* Based on the results obtained from RT-PCR experiments using various reverse primers (as depicted in S1 Fig), the 3'-UTR length of the mRNA transcribed from the *rps7b-102* variant in the *tl(caa)Δint\** is estimated to be approximately 1820–2150 nt. This length is 7–13 times longer than that of the wild-type *RPS7B*, which typically ranges from 170–251 nt [20,21], as consistent with the results of northern blotting (Fig 1B, white triangle). Given that yeast mRNAs with 3'-UTRs of 350 nt or longer are preferentially degraded by NMD [22,26], it was reasonable that NMD targets the longer transcript of *RPS7B* in the *tl(caa)Δint\** mutant.

Some paralogous gene pairs encoding a ribosomal protein show interdependent expression [3]. We speculated that reducing *RPS7B* mRNA in the *tl(caa)Δint\** may alter *RPS7A* expression; therefore, we conducted northern analyses of paralogs of *RPS7A* and *RPS7B* mRNAs (Fig 1B). *RPS7A* mRNA was consistently expressed in the wild-type, *tl(caa)Δint\**, and any *upf1Δ* background strains (Fig 1B, top panel, lanes 1–4): the signal ratio of *RPS7A* between the *tl(caa)Δint\** and the wild-type yeasts was 1.07 ± 0.36. We also tested effect of *rps7bΔ* and found that the deletion affected *RPS7A* expression marginally (*rps7bΔ*/WT, 0.69 ± 0.13; by Student's *t*-test, *p* = 0.0066; n = 3). However, in the case of *RPS7B* mRNA, *rps7aΔ* upregulated *RPS7B* mRNA by nearly 2-fold (*rps7aΔ*/WT, 1.87 ± 0.53; Student *t*-test, *p* = 0.024; n = 3; Fig 1B, middle and bottom panels, lanes 5 and 6). The expression of *RPS7A* appears to dominantly control the Rps7-paralogous expression at the mRNA level for buffering deleterious reduction of itself, whereas that of *RPS7B* has only a marginal role in the paralogous expression.

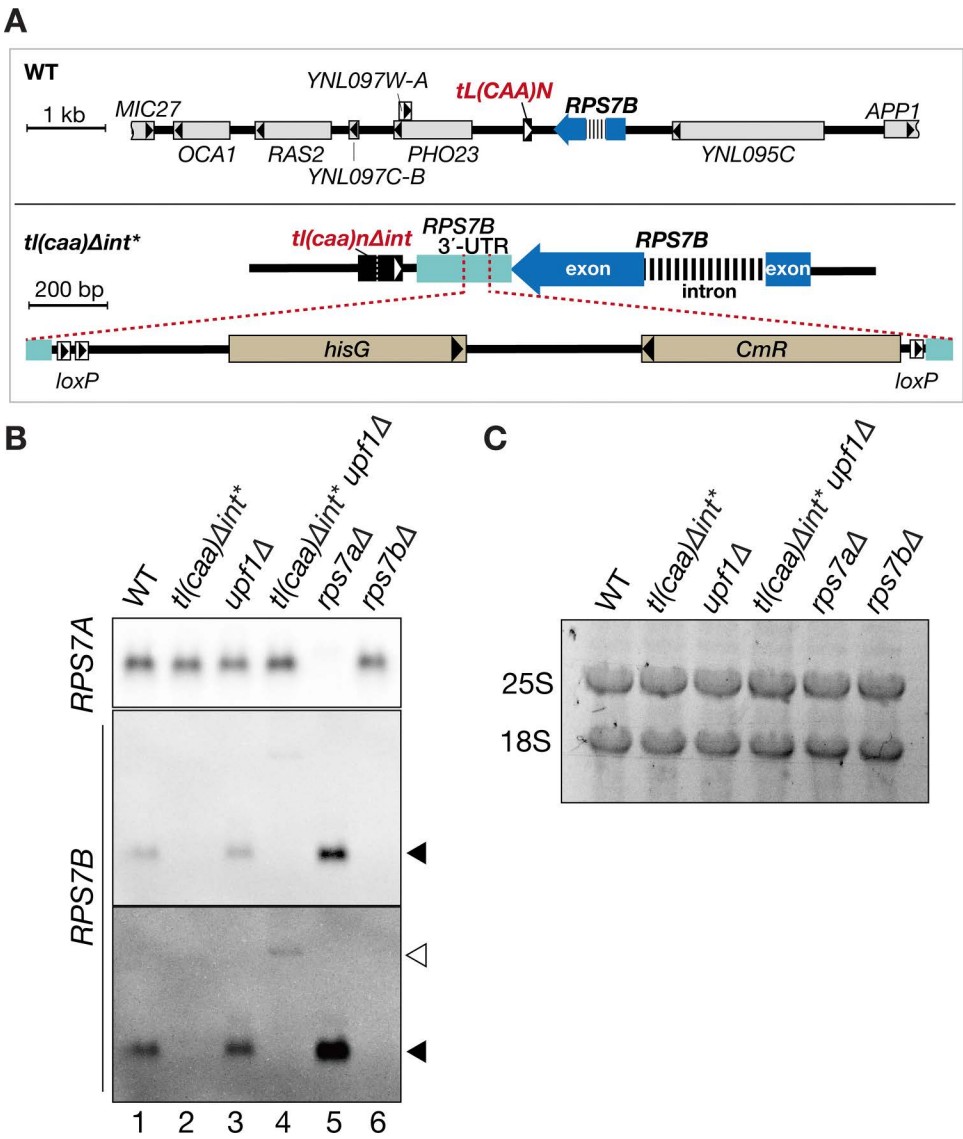

**Fig 1. The *tl(caa)Δint\** has a 2.3-kb insertion in 3'-UTR of *RPS7B* mRNA and its insertion activates NMD for *RPS7B* mRNA. (A)** A 10-kb chromosome region harboring *tL(CAA)N* and *RPS7B* in the wild-type *S. cerevisiae* genome is shown schematically (upper panel). A magnified view of the chromosome region of *tL(CAA)N* and *RPS7B* in the *tl(caa)Δint\** strain (TYSC2148) is also shown (lower panel). The *tl(caa)Δint\** has the 2.3-kb sequence insertion containing *loxP::hisG::CmR::loxP* between *tL(CAA)N* and *RPS7B* genes: *rps7B-102* allele. Black box, *tL(CAA)N*; blue arrow with a thick striped-line insertion, *RPS7B* (the striped line represents intron); gray box, yeast ORFs other than *RPS7B*; beige box, marker genes. A black or white triangle in a box indicates the direction of transcription. In the lower panel, the 3'-UTR of *RPS7B* is highlighted in light blue. Gene names of *tL(CAA)N* and *RPS7B* genes are in italicized bold face in red and black, respectively. **(B)** Northern analyses of *RPS7A* and *RPS7B* mRNAs. Each lane received an RNA sample (5 μg) prepared from the wild-type strain (WT) or the indicated-mutant strains grown at 30°C in YPD. For *RPS7B* mRNA, light-contrast (upper) and dark-contrast (lower) images of the same blot are displayed. Black triangles indicate WT *RPS7A* or *RPA7B* mRNAs. A white triangle indicates a longer transcript of *RPS7B* in the *tl(caa)Δint\** strain. **(C)** Total RNAs in an agarose gel prior to northern analysis were visualized by GelRed staining.

## The addition of a C-terminal HA-tag to *RPS7A* gene, along with the *ADH1* 3'-UTR and terminator, decreased the levels of *RPS7A* mRNA and compromised cell viability, especially when *RPS7B* was defective

Paralogous RP gene expression is regulated by multiple layers of transcriptional and post-transcriptional levels [3]. In try-panosomatids, recent findings show that the translation of RP genes can be regulated by RNA-binding proteins interacting

with their 5'- or 3'-UTRs [27], contributing to paralog-specific expression [28]. Similarly, in yeast, UTRs may serve as regulatory elements for RP paralog expression, potentially playing a more significant role than previously anticipated. To further explore expression difference between *RPS7* paralogs, especially those with alteration in their 3′-UTRs, we modified *RPS7A* mRNA by introducing a C-terminal HA-tag and replacing its 3'-UTR with the *ADH1* 3'-UTR and terminator followed by a *CgHIS3* marker sequence (*RPS7A-HA::CgHIS3*). Although this alternation of *RPS7A* did not inhibit cell growth in the wild-type cell, the introduction of *RPS7A-HA::CgHIS3* in the *tl(caa)Δint\** background diminished growth (Fig 2B, WT *RPS7A-HA::CgHIS3* and *tl(caa)Δint\* RPS7A-HA::CgHIS3*). This growth phenotype that appeared in the *tl(caa)Δint\* RPS7A-HA::CgHIS3* was alleviated by an additional *UPF1* deletion (Fig 2B, *tl(caa)Δint\* upf1Δ RPS7A-HA::CgHIS3*). The relative levels of *RPS7A-HA::CgHIS3* mRNA, which is longer than wild-type *RPS7A*, were approximately 70–80% lower than those of wild-type *RPS7A* across all backgrounds (Fig 2C). Regarding protein levels, all *RPS7A-HA::CgHIS3* strains produced almost equal amounts of Rps7Ap-HA (Fig 2D; one-way ANOVA, $p = 0.66$). Considering the diminished mRNA expression of both *RPS7* paralogs in the *RPS7A-HA::CgHIS3* strains, it appears that *tl(caa)Δint\* RPS7A-HA::CgHIS3* yields insufficient levels of total Rps7p (Fig 2B–2D). Of note, because *upf1Δ* did not rescue the phenotype of *RPS7A-HA::CgHIS3* (Fig 2C and 2D), NMD was not the reason of decreased mRNA expression from the modified *RPS7A* gene. Collectively, the 3'-UTR of *RPS7A* seems to have some *cis* element(s) to enhance the basal transcription and/or stability of *RPS7A* mRNA.

## The 3'-UTR of *RPS7B* mRNA, derived from the *tl(caa)Δint\**, but not the *CgHIS3* sequence, triggers NMD for *RPS7B* in the wild-type strain

The presence or absence of an intron in a tRNA gene influences its ability to function as an insulator, preventing the spread of silenced chromatin and thereby altering the silencing status of nearby pol II genes [29–31]. Thus, it is possible that the tRNA intron in *tL(CAA)N* additionally affect *RPS7B* expression in the *tl(caa)Δint\** strain. To verify whether only the *loxP::hisG::CmR::loxP* insertion into the *RPS7B* mRNA 3'-UTR, but not the intron removal from the *tL(CAA)* genes, resulted in the observed phenotypes, we generated two types of *rps7b* derivatives, *rps7b-101* and *rps7b-102*, which possess different insertions within the 3'-UTR of *RPS7B* at the same position as the *tl(caa)Δint\** mutant, and introduced into strains with the wild-type *tL(CAA)* genes. In the *rps7b-101* strain, a *CgHIS3* marker was introduced as an inserted sequence. In contrast, the *rps7b-102* strain carries the *rps7b-102* allele, which contains the *rps7b::loxP::CmR::hisG::loxP* insertion retrieved from the *tl(caa)Δint\** strain (Fig 3A). Unlike *tl(caa)Δint\**, neither *rps7b-101* nor *rps7b-102* alone conferred any growth defect at 30°C on YPD (Fig 3B). Northern analyses revealed that the authentic *RPS7B* mRNA was barely detected in the *rps7b-102* mutant while the longer transcript (white triangle) was stabilized by introducing *upf1Δ* like the *tl(caa)Δint\** mutant (Fig 3C, lanes 3 and 7). Consequently, the *rps7b-102* recapitulated the phenotypic characteristics of *tl(caa)Δint\** in terms of *RPS7B* mRNA expression, which is subject to NMD, and these phenotypes remained unaffected by the presence or absence of intron in the tRNA$^{Leu}_{CAA}$ genes. We further examined the effects of very low-level expression of *RPS7B* in the *rps7b-102*, possessing the *RPS7A-HA::CgHIS3* background. Again, the synthetic growth defect observed in the *tl(caa)Δint\** was reproduced in the *rps7b-102* derivative, and the defect was suppressed by *upf1Δ* (Fig 3D). Stabilization of *rps7b-102* mRNA by *upf1Δ* in the *RPS7A-HA::CgHIS3* background was confirmed by northern blotting (Fig 3C, lanes 9 and 10). RT-PCR analyses were conducted with cDNA obtained from the *rps7b-102* mutant that harbors the wild-type *tL(CAA)* genes and *upf1Δ*. These analyses revealed that the 3'-UTR of the *rps7b-102* mRNA terminates approximately 1820–2150 nt downstream of the stop codon under the wild-type *tL(CAA)* background, mirroring the findings observed in cDNA from the *tl(caa)Δint\** mutant. And, the tRNA intron had no effect on *RPS7B* expression with the long 3'-UTR.

Intriguingly, the *rps7b-101* mutant that has the *CgHIS3* insertion, instead of the *rps7b-102* allele, produced *RPS7B* mRNA with similar length and signal intensity to the wild type. The mutant did not express any longer transcripts stabilized by *upf1Δ* (Fig 3C, lanes 4 and 8). RT-PCR analyses showed that *RPS7B* mRNA in the *rps7b-101* mutant has

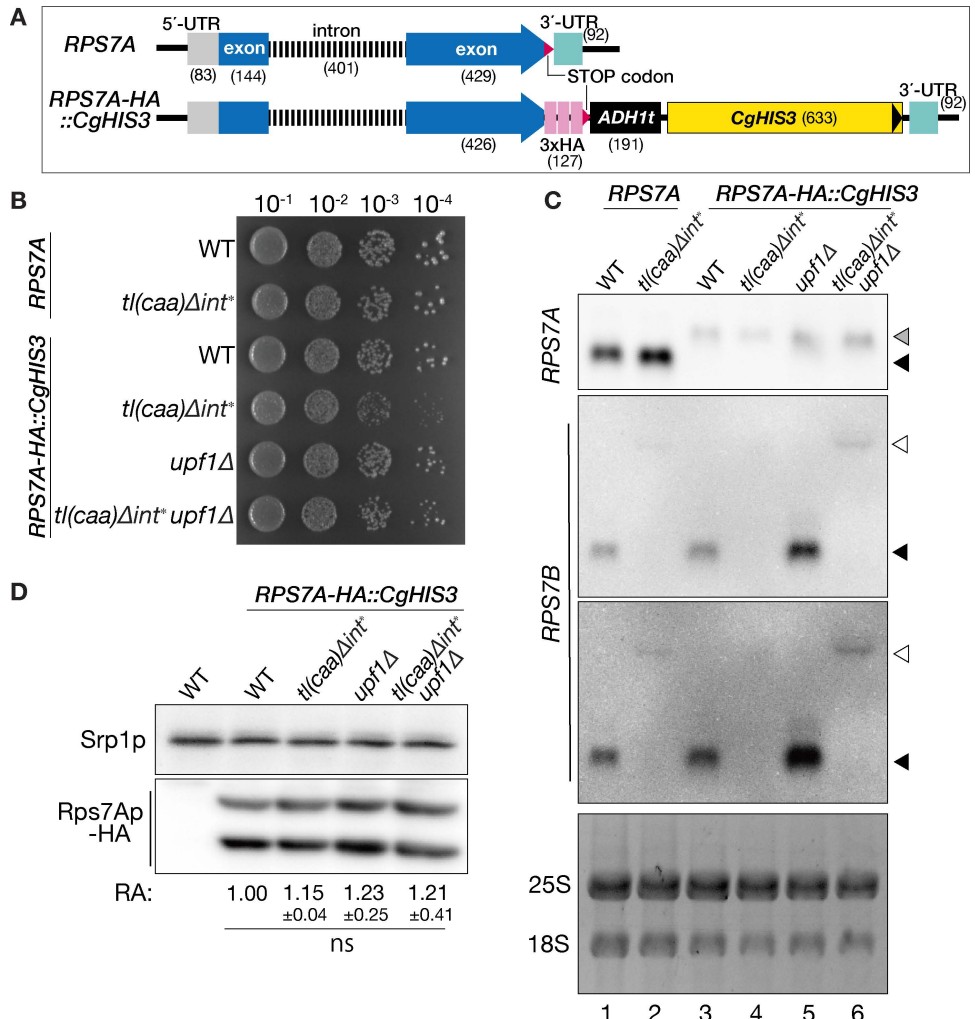

**Fig 2. The C-terminal HA-tagging of the *RPS7A* gene with the *ADH1* terminator reduces *RPS7A* mRNA expression in an NMD-independent manner. (A)** The panel illustrates the WT *RPS7A* and the *RPS7A-HA::CgHIS3* genes. Stop codons are marked in magenta. A number in parentheses represents nucleotide length of the corresponding part (bp). **(B)** Growth comparison on a YPD plate at 30°C. The upper set of wild type (WT) and *tl(caa)Δint\** strains has the wild-type *RPS7A* gene. The lower set of the WT and *tl(caa)Δint\** strains has the C-terminally HA-tagged *RPS7A* gene with the *ADH1* terminator and *CgHIS3* marker (*RPS7A-HA::CgHIS3*) in the presence or absence of the *UPF1* gene. Saturated cultures of the indicated strains were serially diluted by 10-fold as shown in the bottom and dropped onto the plate. **(C)** Northern analyses of *RPS7A* and *RPS7B* mRNAs in the strains used in **(B)**. Cells were grown at 30°C in YPD. Black triangles indicate WT *RPS7A* or *RPS7B* mRNAs. Grey and white triangles indicate *RPS7A-HA* and the longer transcript of *rps7b* in *tl(caa)Δint\**, respectively. For *RPS7B* mRNA, light-contrast (upper) and dark-contrast (lower) images of the same blot are displayed. The bottom panel represents total RNAs visualized by GelRed staining prior to northern analysis. **(D)** Western blot analysis of the *RPS7A-HA::CgHIS3* strains used in **(B)** and **(C)**. Equal amounts of total protein were extracted from the cells, and these extracts were subjected to immunoblotting analysis. An antibody specifically designed to detect the HA tag was used, with anti-Srp1p antibodies serving as a control for loading. Numbers represent relative abundance (RA) of HA-tagged Rps7Ap that normalized by Srp1p in n ≥ 3. The mean value of the WT was set to 1.00. ns: not significant by one-way ANOVA, $p > 0.05$.

its major 3'-terminus between the 206–265 nt downstream the termination codon and its minor terminus between the 265–598 nt (S2 Fig). Indeed, the former major terminus is similar to that of the wild-type (170–251 nt) [20,21]. Therefore, it seems that the inserted *CgHIS3* sequence coincidentally introduces cryptic poly-A signals within a permissive range to avoid NMD.

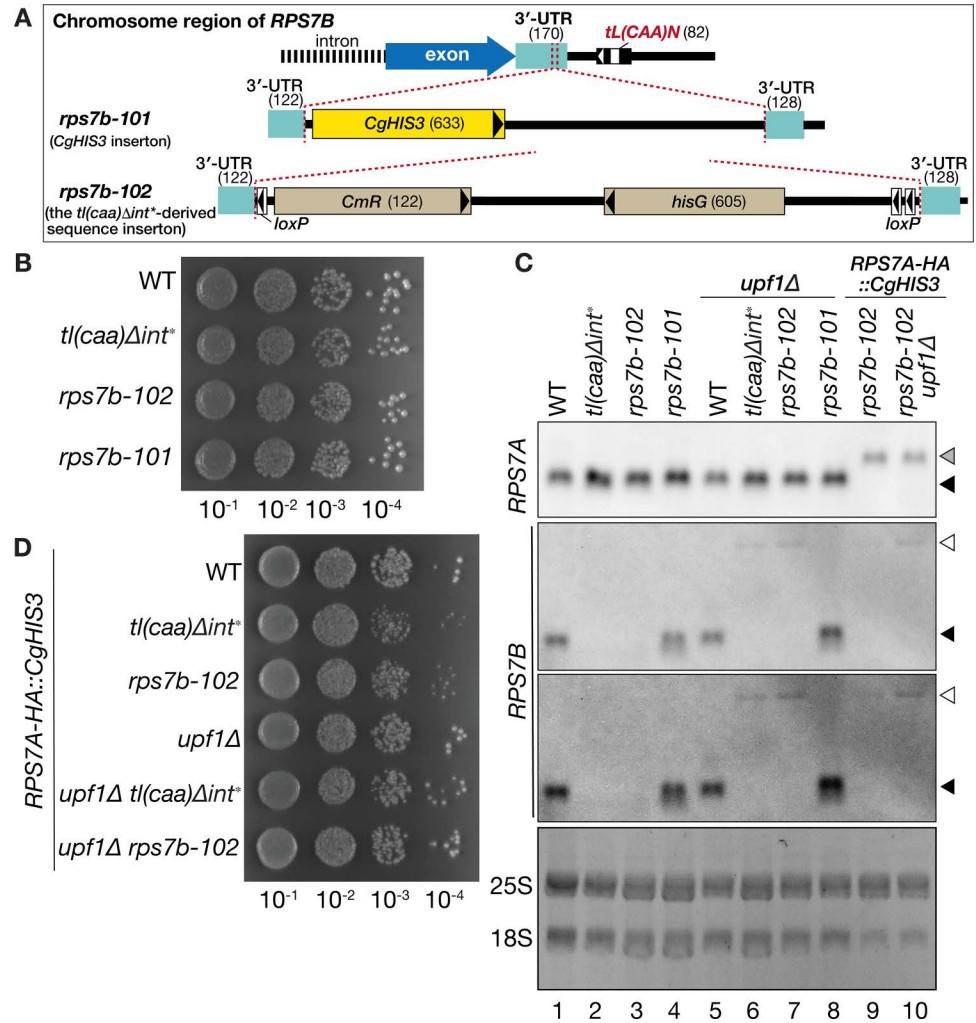

**Fig 3. NMD for *RPS7B* mRNA with altered 3'-UTR is affected by the inserted sequences. (A)** The panel illustrates the 3'-part of *RPS7B* of the wild-type, *rps7b-101* (*CgHIS3* insertion), and *rps7b-102* alleles. A number in parentheses represents nucleotide length of the corresponding part (bp). **(B)** Growth comparison of the wild-type, *tl(caa)Δint\**, *rps7b-102*, and *rps7b-101* strains on a YPD plate at 30°C. **(C)** Northern analyses of *RPS7A* and *RPS7B* mRNAs in the indicated *rps7b-101*- and *rps7b-102*-based strains. Cells were grown at 30°C in YPD. Black triangles indicate WT *RPS7A* or *RPS7B* mRNAs. Grey and white triangles indicate *RPS7A-HA* and the longer transcript of *rps7b-102* mutants, respectively. For *RPS7B* mRNA, light-contrast (upper) and dark-contrast (lower) images of the same blot are displayed. The bottom image is GelRed-stain of the RNA gel as in Fig. 2C. **(D)** Growth comparison of the indicated *RPS7A-HA::CgHIS3* variants on a YPD plate at 30°C.

## Abundance of FLAG-tagged Rps7Bp demonstrates a positive correlation with mRNA abundance: it is upregulated in the *rps7aΔ* strain and downregulated in the *tl(caa)Δint\** strain

According to Ghulam *et al.* [7], Rps7p expression is regulated by both transcriptional and translational levels. To assess the transcriptional effect of *RPS7B*, we first performed semi-quantitative RT-PCR. As shown in S3 Fig, the primary transcript of *RPS7B* having its intronic sequence was not reduced in the *tl(caa)Δint\** nor the *rps7b-102* mutants, implying that the transcriptional process of *RPS7B* gene was not compromised by the marker insertion in its 3'-UTR. We then examined whether the *tl(caa)Δint\** strain impaired *RPS7B* mRNA expression post-transcriptionally, hypothesizing that mRNA abundance directly reflects protein levels due to aberrant 3'-UTR. To analyze whether Rps7Bp expression correlates with

its mRNA abundance, we performed western blotting using strains expressing FLAG-tagged Rps7Bp in the wild-type, the *tl(caa)Δint\** and the *rps7aΔ* backgrounds. The *FLAG* tags were introduced at the end of the ORF, and conjugated with the stop codon and 3'-UTR of *RPS7B* in each strain (Fig 4A). As shown in Fig 4B and 4C, the yield of FLAG-tagged Rps7Bp reduced to one-fourth in the *tl(caa)Δint\** mutant (*tl(cca)Δint\**/WT, 0.26 ± 0.13; pairwise comparison with Benjamini-Hochberg False Discovery Rate (FDR) used as a multiple test correction method, hereafter BH; *p* < 0.0001), but rose more than 1.5 times in the *rps7aΔ* (*rps7aΔ*/WT, 1.71 ± 0.09; BH, *p* < 0.0001). mRNAs encoding *RPS7B-FLAG* variants were less stable compared with the untagged *RPS7B* mRNA (Fig 4D, lanes 1 and 4), The FLAG-tagged mRNAs retained expression difference of their untagged counterparts (Fig 4D, lanes 1–3 versus lanes 4–5). These findings, in conjunction with others (Figs 1 and 2), demonstrate that disruption of proper *RPS7A* expression leads to an enhancement in *RPS7B* mRNA expression, subsequently results in an upregulating of Rps7Bp production in accordance with the mRNA level.

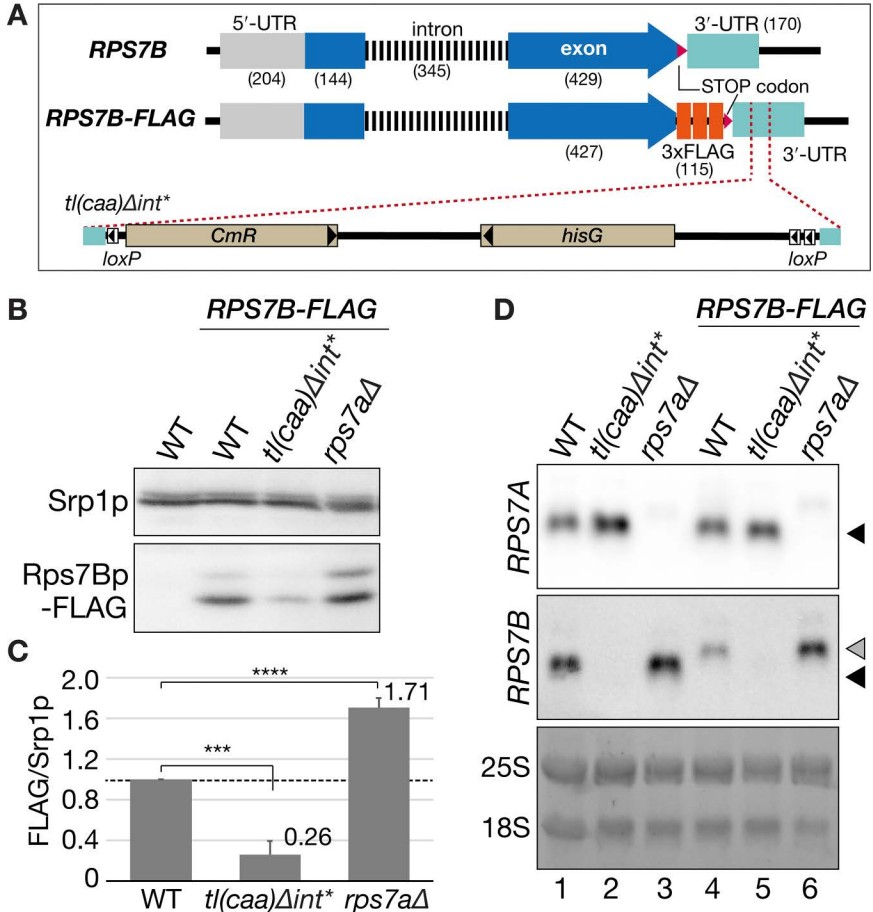

**Fig 4. Translational status of Rps7Bp in the *tl(caa)Δint\** and *rps7aΔ* strains. (A)** The panel illustrates the *RPS7B* region of the wild-type, *RPS7B-FLAG,* and *RPS7B-FLAG* with the *tl(caa)Δint\**-derived insertion alleles. **(B)** Western blot analysis of FLAG-tagged Rps7Bp in the wild type (WT), *tl(caa)Δint\** and *rps7aΔ* background. Total cell extracts were analyzed as in Fig. 2D. **(C)** A bar chart represents relative abundance of FLAG-tagged Rps7Bp quantified and normalized by abundance of Srp1p in **(B)**. Multiple comparisons were performed using the Benjamini-Hochberg (BH) method (****, *p* < 0.0001, n = 3 or 4). The wild-type level (1.0) is shown as a dashed line. Error bars indicate standard deviation. **(D)** Northern analyses of *RPS7A* and *RPS7B* mRNAs in the indicated strains. Cells were grown at 30°C in YPD, and log-phase cells were harvested. Black triangles indicate WT *RPS7A* and *RPS7B* mRNAs. A grey triangle indicates *RPS7B-FLAG* mRNA. The bottom panel represents total RNAs visualized by GelRed staining prior to northern analysis.

## General RP gene transcriptional regulators, Fhl1p and Rap1p, play major roles of *RPS7*-paralog responsive expression

In the traditional view, regulation of RPs often occurs at protein level or in response to the amount of assembled ribosomes, with feedback process frequently leading to transcriptional repression of RP genes [32]. Thus, *RPS7* paralog-specific feedback response would initiate at transcriptional level, and in this process, the upstream and/or downstream regions of the ORF may provide clues to understanding the *RPS7*-specific gene regulation mechanism *in vivo*. To explore difference of transcriptional features between *RPS7A* and *RPS7B* genes, we first focused on the upstream region and searched for known *cis* elements located in the promoter region of *RPS7A* and *RPS7B* on Saccharomyces Genome Database. Indeed, the *RPS7* promoters have consensus sequences for prominent transcription factors, Fhl1p and Rap1p [33] (Fig 5A). To assess impacts of these *cis* elements on *RPS7* expression, we conducted reporter gene assay utilizing both wild-type and mutated versions of their respective promoters and the common *CYC1* terminator (S2 Fig). *RPS7A* contains one binding site each for Rap1p and Fhl1p with a 5-base pair overlap between the two sites. We deleted this entire sequence covering the Rap1p and Fhl1p binding sites to make a mutated version of the *RPS7A* promoter (Fig 5A, *rps7a-mut*). Conversely, *RPS7B* contains two Fhl1p binding sites, one of which overlaps with the Rap1p consensus sequence; we simultaneously deleted these two sites from the *RPS7B* promoter region (*rps7b-mut*).

In wild-type cells (WT), deletion of the Fhl1p and Rap1p *cis* elements from the *RPS7A* promoter (*rps7a-mut*) did not change the reporter expression (Fig 5B; $0.78 \pm 0.22$ vs $0.91 \pm 0.25$, BH within the same *RPS7* genetic background, $p > 0.05$), while the mutated version of the *RPS7B* promoter (*rps7b-mut*) failed to drive the reporter expression ($0.68 \pm 0.25$ vs $0.04 \pm 0.09$; BH within the same *RPS7* genetic background, $p = 0.013$). The Fhl1p- and/or Rap1p-binding sites are selectively crucial for *RPS7B* expression under the WT background. Consistent with the expression difference of endogenous *RPS7B* between WT and *rps7aΔ* strains (Fig 4), the wild-type *RPS7B* promoter led to a 2.9-fold increase in the reporter expression in the *rps7aΔ* compared to that in the WT cells (Fig 5B, BH over different *RPS7* genetic backgrounds, $p = 0.038$). Similarly, the wild-type *RPS7A* promoter also yielded a 7.7-fold increase in reporter expression in the absence of chromosomal *RPS7A* (BH over different *RPS7* genetic backgrounds, $p < 0.0001$). This suggests the presence of a transcriptional feedback mechanism affecting both *RPS7A* and *RPS7B* in response to the level of Rps7Ap. Although this *cis* element mutation on the *RPS7A* promoter did not affect reporter expression in the WT background, the same mutation resulted in reduction of the reporter expression in the *rps7aΔ* ($6.02 \pm 1.55$ vs $2.99 \pm 0.96$; Fig 5B, BH within the same *RPS7* genetic background, $p = 0.0078$). Thus, the *cis* elements for Fhl1p and/or Rap1p are required for full response of *RPS7A* expression in the *rps7aΔ* background. Further, as expected, the *rps7b-mut* promoter could not drive reporter expression in the *rps7aΔ* background. In the *rps7bΔ* cells, both the wild-type *RPS7A* and *RPS7B* promoters led to comparable reporter expression to that in the WT cells (Fig 5B; $1.09 \pm 0.21$ and $1.06 \pm 0.48$, respectively). While disruption of the *cis* element for Fhl1p and Rap1p in the *RPS7A* promoter seemed to reduce reporter expression, this was not statistically significant (Fig 5B; BH within the same *RPS7* genetic background, $p > 0.05$). Again, no reporter expression was seen with the *rps7b-mut* promoter in the *rps7bΔ*. Collectively, the upstream region, particularly the Fhl1p and/or Rap1p binding sites, play a crucial role in *RPS7* paralog specific response.

Secondly, to evaluate the impact of the downstream region on their paralog-specific expressions, we additionally generated reporter genes in which constructs share a common *ADH1* promoter and *GFP-FLAG-HIS3* gene, with the only difference being the 3'-UTRs and terminators (*RPS7A* or *RPS7B*), in contrast to the upstream region. As shown in Fig 5C, although the overall expression of these reporter genes was mildly decreased in the *rps7aΔ* background, either cases that carry the *RPS7A* or *RPS7B* 3'-UTR and terminator showed similar reporter expression among wild-type, *rps7aΔ*, and *rps7bΔ* strains. Thus, the 3'-UTRs of *RPS7* paralogs alone had a limited impact on paralog-specific gene regulation, prompting us to further investigate the role of the promoter regions.

Since Fhl1p and/or Rap1p binding sites are crucial for *RPS7*-paralog specific response (Fig 5B), we generated an *fhl1Δ* strain and repeated the reporter assay to analyze whether Fhl1p acts on *RPS7* paralog expression. Because *RAP1*

 

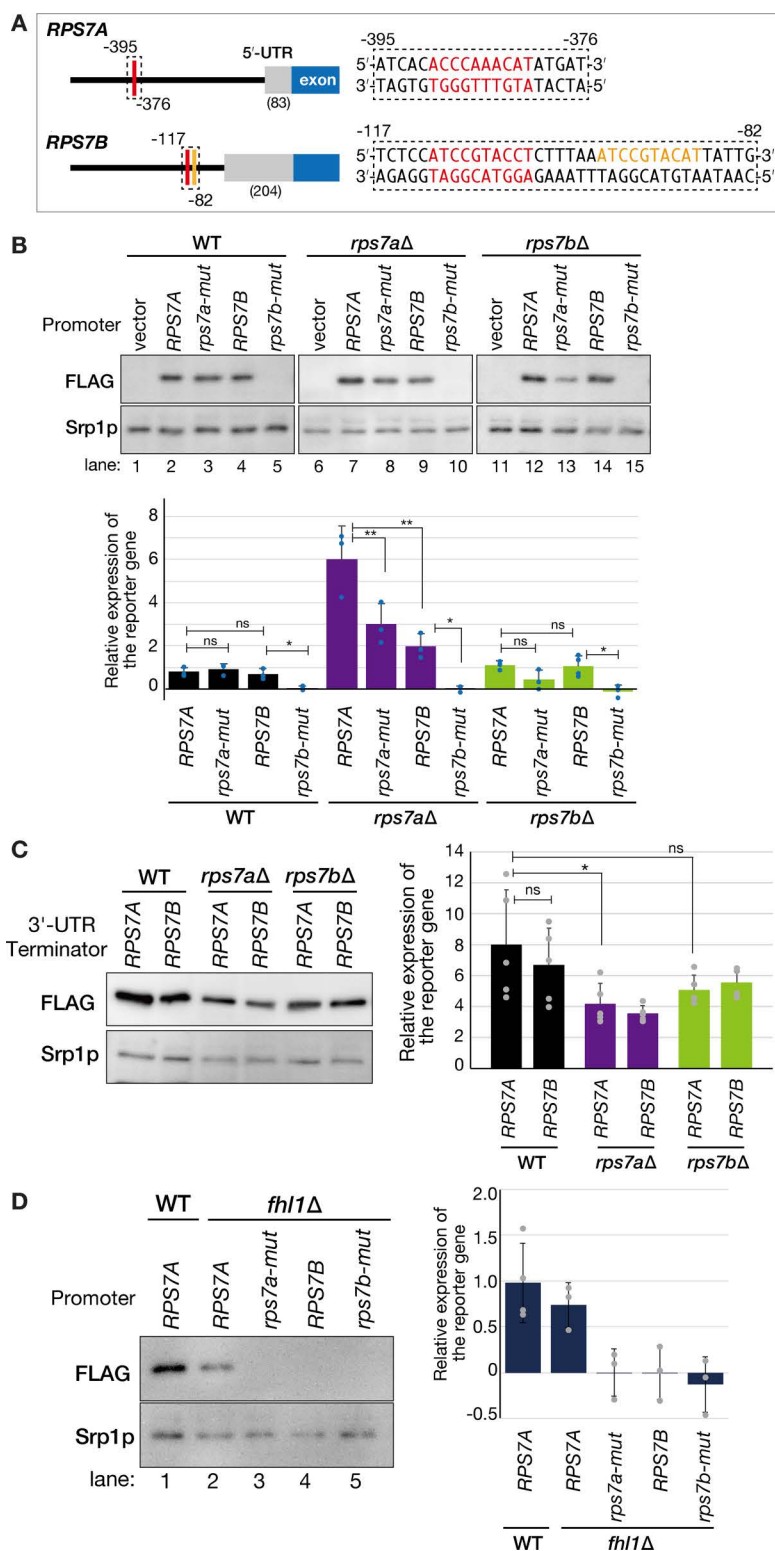

**Fig 5. The transcriptional response of *RPS7* paralogs relies on Rap1p and Fhl1p binding. (A)** The left panel illustrates the upstream regions of *RPS7A* and *RPS7B* genes. Areas where Rap1p and Fhl1p bind simultaneously are highlighted in red, while a region where Fhl1p bind alone is marked in orange. A number in parentheses represents nucleotide length of the 5'-UTRs (bp). The right panel shows enlarged sequences of the dotted line areas

from the left panel. The numbers represent the upstream region relative to the transcription start site. The simultaneous binding sites for Rap1p (*RPS7A*: 5'-ATGTTTGGGT-3'; *RPS7B*: 5'-AGGTACGGAT-3') and Fhl1p (*RPS7A*: 5'-ACCCAAACAT-3'; *RPS7B*: 5'-ATCCGTACCT-3') are highlighted in red, while the Fhl1p binding site alone in *RPS7B* (5'-ATCCGTACAT-3') is shown in orange. **(B)** Western blot analyses to assess expression of the *GFP-FLAG-HIS3* reporter gene under the control of wild-type or mutated *RPS7* paralogous promoters in the indicated strains. The mutated versions of the promoters (*rps7a-mut* and *rps7b-mut*) lack the Fhl1p and/or Rap1p binding sites. Total cell extracts were analyzed as in Fig 2D. A bar chart represents the relative abundance of the reporter-gene product normalized by the abundance of Srp1p across at least three biological replicates. Multiple comparisons were performed using with the BH method (ns, not significant; *, $p < 0.05$; **, $p < 0.01$, n = 3 or 4). Error bars indicate standard deviation. **(C)** Western blot analyses of the *GFP-FLAG-HIS3* reporter gene driven by the *ADH1* promoter and harboring 3'-UTR of *RPS7A* or *RPS7B* gene in the indicated strains. Total cell extracts were analyzed as in Fig 2D. A bar chart represents the relative abundance of the reporter-gene product normalized by the abundance of Srp1p in more than three biological replicates. Multiple comparisons were performed using with the BH method (ns, not significant; *, $p < 0.05$, n = 4 or 5). **(D)** Western blot analyses of the *GFP-FLAG-HIS3* reporter gene described in (B) in the wild-type and *fhl1Δ* strains. A bar chart represents the relative abundance of the reporter-gene product normalized and presented as in (B).

is essential for yeast growth we could not conduct deletion analysis of this transcription factor. As shown in Fig 5D, *fhl1Δ* did not lose reporter expression driven by the wild-type *RPS7A* promoter (lane 2), while the expression was completely abolished in the other cases (lanes 3–5). Considering that the wild-type *RPS7A* promoter was still active in the *fhl1Δ* strain (Fig 5D, lane 2) and the *rps7a-mut* promoter functioned in the WT cells (Fig 5B, lane 3), the Fhl1p-site in the *RPS7A* promoter region may be recognized not only by Fhl1p but also by another factor; probably Rap1p, and Fhl1p may bind an additional site in this region or via interaction with other DNA-binding proteins. Conversely, transcription of *RPS7B* strictly depends on Fhl1p, suggesting that Fhl1p drives its expression by binding at least one of the two Fhl1p-binding sites of the *RPS7B* promoter. Thus, the different contributions of Fhl1p (and/or Rap1p) to the *RPS7*-paralog promoter regions are key determinants of *RPS7*-paralog expression, which is influenced by the levels of Rps7Ap (or total Rps7p).

## Discussion

Our study revealed an interesting finding: the deletion of *RPS7A* led to an upregulation in the expression of *RPS7B*, as shown in Fig 1B. However, the opposite effect was not observed; deleting *RPS7B* did not reciprocally impact the expression of *RPS7A*. This observation suggests that *RPS7A*, but not *RPS7B*, works as an expression modulator of its counterpart (Fig 1B). Our serendipitous finding began with an unexpected genomic change near a tRNA gene during construction of the *tl(caa)Δint* strain. Although our case is completely artificial, insertion are often introduced at genomic fragile sites by transposons [34]. Indeed, tRNA loci exhibit mutation rates approximately 7–10 times greater than the genome-wide average [35]. Thus, genomic changes or mutations similar to our mutant might arise under natural conditions during evolution.

Asymmetrical regulation has been described in several RP-paralog pairs [3]. It is considered that not only transcriptional initiation but also transcriptional termination and/or splicing acts as the decisional step defining amounts of the RP mRNAs, and rather RNA degradation is not a major determinant of RP mRNA abundance [3]. Since the production of FLAG-tagged Rps7Bp was well correlated with its mRNA level (Fig 4B–4D), the critical regulatory step in Rps7Bp expression driven by *RPS7A* products seems to rely on mRNA abundance. Indeed, our reporter assay revealed that reporter expression from the *RPS7B* promoter is upregulated by deletion of the *RPS7A* gene, but also indicated that the *RPS7A* promoter is also under feedback regulation. Additionally, well-known transcription factors of RP genes, Fhl1p and/or Rap1p, contributed to the full activation of *RPS7A* and *RPS7B* (Fig 5B and 5C). Notably, the *RPS7B* but not *RSP7A* promoter-driven expression was completely abolished when *FHL1* was deleted (Fig 5D and 5E; see below). These results showed asymmetric features of *RPS7* paralogous genes in transcriptional regulation. In addition, we found that transcription of *RPS7A* is also under the feedback regulation by Rps7Ap, suggesting that maintaining total level of Rps7p/eS7 is important for yeast physiology. Despite of such regulatory mechanism, deletion of one of RP paralog pairs can lead pivotal effects on growth competition among yeast cells, which sometimes result in survival of mutants with drastic genomic alterations, such as aneuploidy [4,8]. Therefore, in the case of Rps7p regulation, the increase in *RPS7B* mRNA levels that observed in *rps7aΔ* is possibly due to aneuploidy, specifically a duplication of the chromosome that carries *RPS7B*.

The *rps7b-102* mutation found in the *tl(caa)Δint\** strain resulted in generation of *RPS7B* mRNA with an approximately 2-knt 3'-UTR, subsequently triggering its degradation through NMD (Fig 1B and S1 Fig). This occurs irrespective of the presence or absence of the intron in the *tL(CAA)* loci (Fig 3C). It is well-established that the majority of transcripts with 3'-UTRs of 350 nt or longer are subject to degradation via NMD in the yeast [22,26], while there are some exceptions, such as *SSY5*, which possess 3'-UTRs ranging in size from 420 to 500 nt [22]. In the case of the *rps7B-101*, cells utilized some multiple cryptic transcription termination sites for *RPS7B* mRNA (S2 Fig). Due to the primary termination occurring approximately 200–270 nt downstream the stop codon, which is short enough to escape from NMD, the predominant *rps7b-101* transcript exhibited a similar expression level to the wild-type (Fig 3C). So far, we have been unable to identify conventional polyadenylation signal motifs, such as positioning elements AAUAAA/AAAAAA or upstream elements UAU-AUA [36], in close proximity to the anticipated 3'-ends of the 3'-UTRs of *rps7b-102* or *rps7b-101*. The mechanism by which yeast cells select these sites for poly-A addition remains unclear.

In the case of HA-tagged *RPS7A* mRNAs, an *ADH1* terminator was utilized instead of the authentic one in *RPS7A*, resulting in lower expression compared to the wild type (Fig 2C). Since the deletion of *UPF1* did not rescue this phenotype, it is conceivable that HA-tagged *RPS7A* mRNAs may be susceptible to transcriptional repression and/or degraded mediated by an mRNA degradation pathway(s) distinct from NMD. Indeed, the RP-paralog pairs generally have heterogeneous 3'-UTRs. Specifically, the minor paralog tends to exhibit alternative termination sites and/or a longer 3'-UTR [3]. Therefore, alteration of the 3'-UTR could have a significant impact on the abundance of *RPS7* mRNAs by potentially losing some *cis*-elements that are unique to RP mRNAs. We did not investigate the effect of a 3'-UTR insertion in the *RPS7A* 3'-UTR, specifically a 2.3-kb insertion of *loxP::CmR::hisG::loxP*, as seen in *rps7b-102*. Given the genomic context, there are no tRNA genes or fragile sites around *RPS7A*, suggesting that natural insertion would not occur. However, more comprehensive studies are needed to examine the impact of 3'-UTR insertions in *RPS7A* to better understand the overall imbalance in regulation between paralogous genes.

Since FLAG-tagged *RPS7B* mRNAs, utilizing the authentic promoter, displayed reduced expression even with the *RPS7B-FLAG* gene featuring the genuine 3'-UTR (Fig 4D), it appears that the insertion of an approximately 100 bp fragment into the C-terminal part of the ORF exerts a significant influence on *RPS7B* expression. Codon utilization and other ORF features are expected to coevolve with mRNA properties, such as length, translational efficiency, and 3'-UTR regulation; consequently, all of these factors, individually or in concert, have potential to impact mRNA stability [37–39]. There may be interdependence between a regulatory sequence(s) in the 3'-UTR and its co-acting element(s) in the ORF of *RPS7B*. Thus, alteration in length between the ORF and the 3'-UTR may affect stability of *RPS7B* mRNA.

Our reporter gene assay showed the different contributions of Fhl1p and/or Rap1p on *RPS7* paralogs expression, and also showed that *RPS7A/B* regulation according to the Rps7Ap level is mainly carried out via their promoter regions but not via their 3'-UTRs. The *RPS7A* promoter is under feedback regulation by abundance of Rps7Ap (or sum of Rps7 proteins), and this regulation is partially compromised by the Fhl1p site deletion. Fhl1p is a RP gene-specific transcription factor collaborating with an essential transcription activator Ifh1p, and has a pivotal role in RP regulation upon various stresses, including unbalanced expression of RPs [40,41]. Deletion of the Fhl1p site in the *RPS7A* promoter had the least effect on reporter expression, and the *FHL1* gene deletion alone has a marginal effect on the *RSP7A/B* promoter activity. However, simultaneous deletion of these two elements completely abolished the reporter expression. This suggests that the Fhl1p site in the *RPS7A* promoter is dispensable for Fhl1p function, as Fhl1p can still be recruited to the *RPS7A* promoter via interactions with other transcription factors, such as Rap1p (a pioneer transcription factor [42]) and Sfp1p (a transcription factor requited by Rap1p [40]), and/or Hmo1p (an HMG family chromatin interactor [43]), which associated with the *RPS7A* promoter region by themselves [40,43,44]. Indeed, it was reported that *IFH1* but not *FHL1* is essential for yeast growth [41], and that a DNA-recognizing fork-head domain is not crucial for Fhl1p function [45]. Thus, in the absence of Fhl1p, another factor bound to the Fhl1p site may recruit Ifh1p to the *RPS7A* promoter. If

this Fhl1p-independent recruitment of Ifh1p to the *RPS7A* promoter requires the Fhl1p site or a surrounding sequence, concomitant deletion of the Fhl1p site and the *FHL1* gene may hinder Ifh1p from associating with the *RPS7A* promoter and its transcription. On the other hand, both the Fhl1p site and Fhl1 protein are indispensable to transcription from the *RPS7B* promoter. Although *RPS7A* and *RPS7B* belong to the same Category I RP genes, where Hmo1p contributes to recruitment of Fhl1p•Ifh1p to their promoter [46], the *RPS7A* promoter does not require Fpr1p, an FKBP12 homolog in the yeast, for its full function, unlike the *RPS7B* promoter. This difference in promoter architecture between these paralogs may result in the difference in requirement of the Fhl1p site and Fhl1p itself. Since deletion of either of these genetic elements completely disrupts the *RPS7B* promoter activity, contribution of these elements to Rps7Ap-dependent regulation of the *RPS7B* promoter is still to be analyzed. Moreover, Rps7Ap may directly interact with a transcriptional regulator to achieve *RPS7A*-specific feedback regulation, as known by the general autoregulatory mechanism of RP production [3]. At least, Fhl1p is not the target of Rps7Ap (Fig 5), indicating that Rap1p is the prime target identified so far. While Rap1p is essential, the use of a *ts*-mutant [47,48] or Auxin-degron system, and the *rap1Δsil* mutant [49] offers valuable tool for further investigation.

Eukaryotic specific ribosomal protein Rps7p/eS7 locates at the 40S head region contacting with Rps13p/uS15, Rps22p/uS8 and Rps27p/eS27 [2,50]. In humans, Rps7p is one of the RPs have implicated in hematopoietic disorders, most notably Diamond-Blackfan anemia [51]. A splicing variant of the human *RPS7* 5'-UTR decreases mRNA level followed by protein reduction, implying that the *RPS7* expression level itself is a pathological cause [52]. In budding yeast, *RPS7A* is a major paralog regarding mRNA expression, reflecting RNA polymerase II association to the ORF, post-cap stability of mRNA, as well as protein abundance [7]. Curiously, the differential incorporation of ribosomal protein paralogs into active ribosomes showed cells' preference for Rps7Bp-containing ribosomes rather than Rps7Ap-containing ones, indicating a discrepancy between paralog utilization in ribosomes and mRNA/protein abundance of the Rps7p-paralog pair [7]. Also at the post-translational levels, only Rps7Ap is ubiquitinated by Not4p, and that cooperated with RQC machinery to support proteostasis positively [53,54]. Thus indeed, yeast duplicated RP genes are unequally controlled at different levels, including transcription, mRNA stability, translation, RP modification, and response to various stimuli [3,7,12].

Our substantiated evidence suggests that the yeast harbors a surveillance system for *RPS7A* mRNA level, and utilizes this information to modulate expression of only *RPS7B*. So far, mechanisms by which the yeast distinctly perceives abundance of *RPS7A* mRNA in a manner independent of *RSP7B* and the underlying biological meanings for this asymmetric regulatory phenomenon remain uncovered. It is important to note that, in many paralogous RP genes, the surveillance or feedback typically occurs through protein level or in response to the amounts of assembled ribosomes. It is also possible that feedback mechanism exists to regulate the expression of different genes when specific subunits are not properly assembled, particularly if ribosomes become stalled at certain stages of translation due to improper expression of paralogous RPs. Comprehensive studies that include both mRNA and protein levels should be considered in future research. Future continuous investigations will unveil more details of these regulatory processes.

## Supporting information

**S1 Fig. RT-PCR of the 3' regions of *rps7b-102* mRNA. (A)** Three prime parts of the *rps7b-102* gene and positions of primers used in the PCR (small red arrows) were schematically shown. **(B)** RT-PCR products of the *rps7b-102* cDNA from SHSC0386 (*rps7b-102 tl(caa)Δint upf1Δ*) amplified by three different cycle numbers were analyzed by agarose gel electrophoresis. Reverse primers used in the PCR are shown on the top of the gel image, and correspond to those in **(A)**. The rightmost three lanes are PCR products with genomic DNA from SHSC0386 as a template, and are used to confirm ability of primers.
(TIF)

**S2 Fig. RT-PCR of the 3' regions of *rps7b-101* mRNA. (A)** Three prime parts of the *rps7b-101* gene and positions of primers used in the PCR (small red arrows) were schematically shown. **(B)** The 3' parts of the *rps7b-101* were amplified from SHSC0388 (*rps7b-101*) cDNA and were analyzed as in S1 Fig. Primers used in the PCR are shown on the top of the gel image, and correspond to those in (A). The rightmost four lanes are PCR products with genomic DNA from SHSC0388 as a template, and are used to confirm ability of primers.
(TIF)

**S3 Fig. RT-PCR of unspliced and spliced forms of *RPS7A* and *RPS7B*. (A)** Schematic drawings of primers (red arrows) used for RT-PCR of *RPS7A* and *RPS7B* mRNAs. The F1 primer of *RPS7A* and the R3 primer of *RPS7B* are designed to hybridize with only spliced mRNAs, and the F2 of *RPS7A* and R2 of *RPS7B* are against intronic regions. **(B)** Gel electrophoresis of RT-PCR samples. cDNAs were synthesized from total RNAs of the strains shown in the bottom list as in S1 Fig except that random hexamers instead of oligo $dT_{20}$ were used as RT primers. Left panels are RT-PCR of the unspliced form of *RPS7A* mRNA (upper) and *RPS7B* mRNA (lower). Right panels are those of their spliced form. In both cases, two PCR cycles (34 and 32 cycles for the unspliced form, and 27 and 25 cycles for the spliced form) are adopted. A primer set used in each amplification is shown under the gel image in red. Lane 1, wild type; lane 2, *tl(caa)Δint\**; lane 3, *rps7b-102*. Details of the primers are described in S4 Table in S1 File.
(TIF)

**S1 File. S1-S4 Tables. Strains and primers used in this study**. S1 Table. Yeast strains used in this study. S2 Table. Primers used to construct stains and plasmids. S3 Table. Plasmid used to construct strains. S4 Table. Primers used for RT-PCR.
(XLSX)

**S2 File. Raw_images.**
(PDF)

## Acknowledgments

We are grateful to Prof. Toshifumi Inada from The University of Tokyo, Japan, for generously providing the pSA144 plasmid for the reporter gene assay.

## Author contributions

**Conceptualization:** Sachiko Hayashi, Tohru Yoshihisa.

**Data curation:** Sachiko Hayashi, Tohru Yoshihisa.

**Formal analysis:** Sachiko Hayashi, Tohru Yoshihisa.

**Funding acquisition:** Sachiko Hayashi, Tohru Yoshihisa.

**Investigation:** Sachiko Hayashi, Tohru Yoshihisa.

**Project administration:** Sachiko Hayashi.

**Resources:** Sachiko Hayashi, Tohru Yoshihisa.

**Supervision:** Sachiko Hayashi, Tohru Yoshihisa.

**Validation:** Sachiko Hayashi, Tohru Yoshihisa.

**Visualization:** Sachiko Hayashi, Tohru Yoshihisa.

**Writing – original draft:** Sachiko Hayashi, Tohru Yoshihisa.

**Writing – review & editing:** Sachiko Hayashi, Tohru Yoshihisa.

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
