## [Decision Letter · Decision Letter 0]

14 Feb 2025

PONE-D-25-01723Beyond the ORF: Paralog-specific regulation of RPS7/eS7 mRNAs via 3'-UTRs and promoter sequences

PLOS ONE

Dear Dr. Hayashi,

Thank you for submitting your manuscript to PLOS ONE. After careful consideration, we feel that it has merit but does not fully meet PLOS ONE’s publication criteria as it currently stands. Therefore, we invite you to submit a revised version of the manuscript that addresses the points raised during the review process.

I have now received the reviewers' comments, and based on their feedback, I request that you revise the manuscript accordingly.

I would like to draw your attention to the comments from Reviewer 2, which include critical points regarding the limitations of the experiments, the conclusions drawn, and the statistical methods applied.

Please submit a revised manuscript after addressing these comments. Additionally, ensure that your revision includes a discussion of any potential limitations in the experimental design or study.

We look forward to receiving your revised manuscript.

Kind regards,

Koppolu Raja Rajesh Kumar, PhD

Academic Editor

PLOS ONE

Journal Requirements:

“This work was supported by JSPS KAKENHI (https://www.jsps.go.jp/english/) Grant number JP20K06491 (to S. H), 17H05672 and 23K18100 (to T. Y.), Takeda Science Foundation (https://www.takeda-sci.or.jp/en/), and Hyogo Science and Technology Association (https://hyogosta.jp/), Japan (to S. H.; 6082). The funders had no role in study design, data collection and analysis, decision to publish, or preparation of the manuscript.”

Reviewers' comments:

Reviewer's Responses to Questions

**Comments to the Author**

1. Is the manuscript technically sound, and do the data support the conclusions?

Reviewer #1: Yes

Reviewer #2: Partly

Reviewer #3: Yes

2. Has the statistical analysis been performed appropriately and rigorously? 

Reviewer #1: Yes

Reviewer #2: No

Reviewer #3: Yes

3. Have the authors made all data underlying the findings in their manuscript fully available?

Reviewer #1: Yes

Reviewer #2: No

Reviewer #3: Yes

4. Is the manuscript presented in an intelligible fashion and written in standard English?

Reviewer #1: Yes

Reviewer #2: Yes

Reviewer #3: Yes

5. Review Comments to the Author

Reviewer #1: The work of Hayashi and Yoshihisa is a demonstration of the use of scientific reasoning to investigate the molecular mechanisms of life from a serendipitous finding. The work contains many results, with varying relevance. Some of them serve to interpret the phenotype of the spontaneous mutant, which has little biological relevance, and others focus on the study of a new mechanism of cross-control between paralogous copies of a yeast ribosomal protein.

1) On page 13, line 194, the mRNA of the CaHIS3 insertion version in Fig. 2C is longer than wt. This should be mentioned in the text.

2) On page 13, line 199, the Rps7A protein level in wt is not shown. This is because wt is not HA-tagged, but it would be interesting to have a relative quantification of Rps7A levels in Rps7A-CgHIS3 with respect to those in wt.

3) Page 15, line 228. There are two ... at the end of the line.

4) Page 17, line 268. “we deleted this sequence...”. It is important to know which sequence was deleted. Since there is a putative 5 bp overlap between the Rap1 and Fhl1 binding sites, if the entire sequence is deleted the consequence would be a lack of both binding sites. If only part of it is removed, Rap1 (or Fhl1) may still bind to it. According to line 272 on page 18 both sites were removed but this conflicts with other conclusions given later. See point #5 below.

5) Line 272 states that (Fig. 5B) the lack of both Rap1/Fhl1 binding sites is not affecting Rps7A expression (as demonstrated by the GFP reporter). This is quite surprising because these factors are described as very important for RP gene expression in many previous papers. Here it appears that in the RPS7A gene they are only required for negative feedback of the Rps7 protein on RPS7A transcription. Although the question is discussed below (pages 23-24) it is quite surprising and as no potential mechanism is suggested it leaves the question too unresolved. Is the Rps7 protein interacting with Rap1 (or Fhl1, or another factor) to prevent binding? If so, the experiment shown below (fhl1 mutant) will demonstrate that Fhl1 is NOT the target of Rps7A. As for the function of Rap1, since Rap1 is essential a ts mutant or Auxin-degron can be used to test the function of Rap1 in Rps7A transcription. Alternatively the rap1delta-sil mutant has been used in the context of RP gene expression where it affects the RNA pol II transition between initiation and elongation (doi:10.1371/journal.pgen.1000614).

6) page 24 line 373. RPS7 is RPS7B?

7) In Discussion (page 24) the authors describe the existence of other TFs that can be activating Rps7A promoter, such as Ifh1 or Hmo1. I think that Sfp1 should be included because it another well-known transactivator of many RP genes.

Reviewer #2: The authors focused on a very interesting topic: what does it happen with paralogous genes? how redundant/specific are they? They have focused on a pair of ribosomal proteins (RPS7A and RPS7B). The main problem of the paper is that the main aim of the study, the specific objectives and the hypotheses are never clearly stated upfront. Therefore, the reader does not know if the described experiments make sense in order to achieve that aim and answer those research questions.

In my opinion, some of the presented experiments seem irrelevant in order to understand the relationship between RPS7A and RPS7B. For example, the authors constructed a strain with an artificially long 3'UTR in RPS7B. Then, they demonstrate that NMD controls the expression of that modified RPS7B gene. However, I wonder why this experiment is relevant if this never occurs in natural/biotechnological strains. WT RPS7B has a normal 3'UTR which is probably not targeted by NMD. Therefore, the implications of these results are totally obscure to me.

Moreover, other comparisons do not seem to be absolutely fair. In my opinion, it is very interesting to know how the amount of mRNA and transcript of one of the paralogs is affected when the other paralog is knocked down or overexpressed. Here, the authors have looked at the effect of the RPS7A deletion on RPS7B. They have also looked at the effect of the RPS7B deletion on RPS7A. However, most of the quantitative analyses on RPS7A where done with the long-UTR RPS7B gene. In this experiment, the transcription of RPS7B is not prevented, but the mRNA is mostly degraded and hence never translated. The reciprocal experiment (RPS7A targeted by NMD) is never done. In order to rule out symmetrical effects, the analyses need to comprehensive.

Finally, I am not entirely satisfied by the way that the statistical analyses are performed, reported and interpreted.

1) I did not see information on the number of replicates used in each experiment; n must be reported.

2) t-test is used in most analyses, but this is incorrect because many of those involve multiple comparisons. If all comparisons are of interest, the authors should use ANOVA plus a post hoc test (e.g. Tukey's test). If only comparisons to a single control are of interest, Dunnet's test should be used instead. If not all the comparisons are of interest and there is no common control, the authors should correct the p-values of the t-test for multiple testing (because they are interested in any significant difference, their type-I error is above 0.05).

3) the authors disregarded an interesting result because it does not agree with their main conclusion (RPS7B expression depends on RPS7A expression, but RPS7A expression is not affected by RPS7B). When describing Figure 1B, the authors state "rps7bΔ affected RPS7A expression only slightly (rps7bΔ/WT, 0.70±0.13; by Student’s t test, p = 0.0066)" in lines 177-178. I totally disagree with this statement. First, the difference is either statistically significant or not; in this case, it clearly is (p-value < 0.01). Slightly is not a statistical term. Second, RPS7A expression is clearly depleted in the rps7bΔ strain; it is just 70% of the expression in the WT strain. A 30% depletion is not a slight change. This result should be taken into account when interpreting the rest of analyses.

For all these reasons, I have to recommend that the paper is rejected. I would suggest to the authors that they decided on what is the main aim of their research and then they did a complete rewriting of the paper focusing on achieving that aim.

Reviewer #3: Hayashi and Youshihisa present their results from investigating the regulation of the yeast RPS7A and B paralogs. Because the yeast genome resulted from a whole-genome duplication event, many proteins are duplicated (termed ohnologs). This includes many ribosomal proteins. Previous work has reported that the alternative copies of ribosomal proteins present in the yeast genome have different regulation and, potentially, somewhat different functions. In this manuscript, the authors report similarities and differences in the regulation of the expression for ohnologs RPS7A and RPS7B. In particular, they report that the short 3' UTR of both genes is important for RNA stability, and that the transcriptional regulation of both RPS7A and 7B respond to a decrease in RPS7A expression, suggesting a feedback loop. The authors also describe some differences in the functions of transcription factor binding sites upstream of the two genes. Overall, this is a well performed study that is interesting to the gene regulation and ribosome research communities. I have mostly minor comments that should be addressed:

1. It has been reported that deletions of ribosomal proteins can cause yeast to become aneuploid as a response to loss of the RP copy (see PMID: 30503772 and PMID: 22377630). To elaborate, segments or entire chromosomes carrying the RP copy that is still present will duplicate, as cells that have the "normal" number of copies are more fit than those that lose one copy (even with the aneuploidy). The authors should note this in the introduction and discussion, as it is relevant to their observation that RPS7A deletion leads to an increase in RPS7B mRNA. It is possible that these increases could be due, at least in part, to amplification of the remaining RPS7 DNA copy.

2. Line 73. What does "delicate to" mean? I don't understand.

3. Line 90 "gene region with an amplified" -> "gene region was replaced with an amplified"

4. Line 192 "phenotype appeared" -> "phenotype that appeared"

5. Line 214 to 215 is confusing. Perhaps try, "Unlike tl(caa)deltaint*, neither rps7b-101 nor rps7b-102 conferred any growth defect at 30C on YPD.

6. Line 273 "unchanged" -> "did not change"

7. Line 304, is that a period after strain? should be comma?

8. Lines 325-326 are confusing.

9. Line 360 "supposed" -> "expected"

10. Line 390 "dependet" -> "dependent"

11. Line 399 " preference for RPS7Bp". The paper referenced shows a preference for RPS7A, not B in higher protein level.

12. Line 406 "demonstrates" -> "suggests". demonstrates is a bit too strong.

13. Line 407 "mRNA level" -> the surveillance or feedback is probably on the protein level or in response to the amount of assembled ribosomes. I don't think there's any evidence that the actual mRNA levels is directly sensed (again why I think it's not "demonstrated"; see comment 12).

14. RPS7A mRNA (probably protein, not mRNA). Most likely mechanism is a problem in ribosome assembly, with ribosomes getting stuck in particular stages of assembly. I wonder if there are feedback mechanisms that turn on expression of different genes when different subunits are not assembled well.

6. PLOS authors have the option to publish the peer review history of their article (what does this mean? ). If published, this will include your full peer review and any attached files.

**Do you want your identity to be public for this peer review?** For information about this choice, including consent withdrawal, please see our Privacy Policy .

Reviewer #1: No

Reviewer #2: No

Reviewer #3: No

---

## [Author Response · Author response to Decision Letter 1]

31 Mar 2025

Point-to-point answers

5. Review Comments to the Author

Important note:

Please refer to the page and line numbers indicated in “Revised Manuscript with Track Changes.pdf” for review. This file includes track changes and highlights in yellow the sections pointed out by the reviewers.

Reviewer #1: The work of Hayashi and Yoshihisa is a demonstration of the use of scientific reasoning to investigate the molecular mechanisms of life from a serendipitous finding. The work contains many results, with varying relevance. Some of them serve to interpret the phenotype of the spontaneous mutant, which has little biological relevance, and others focus on the study of a new mechanism of cross-control between paralogous copies of a yeast ribosomal protein.

As you pointed out, we originally interested in tRNA biology and began the manuscript with our unexpected (artificial) mutant and the observation made from this mutant. While the biological relevance of our mutant is limited, considering that tRNA gene regions are often targeted by transposon (Guimarães, et al., 2021, Front. Microbiol., 12, 634004, doi: 10.3389/fmicb.2021.634004), such genomic changes altering the 3'-UTR of RPS7B could occur during evolution (and were probably eliminated by natural selection). Therefore, we believe our serendipitous finding holds significance, at some level, within the context of the manuscript. At this point, we added this to the discussion. Please refer to the following.

Page 22, lines 426-431:

Our serendipitous finding began with an unexpected genomic change near a tRNA gene during construction of the tl(caa)∆int strain. Although our case is completely artificial, insertion are often introduced at genomic fragile sites by transposons [34]. Indeed, tRNA loci exhibit mutation rates approximately 7 to 10 times greater than the genome-wide average [35]. Thus, genomic changes or mutations similar to our mutant might arise under natural conditions during evolution.

1) On page 13, line 194, the mRNA of the CaHIS3 insertion version in Fig. 2C is longer than wt. This should be mentioned in the text.

Thank you for your advice. We revised the sentence, including the corresponding part as below.

Page 13, lines 231-233:

The relative levels of RPS7A-HA::CgHIS3 mRNA, which is longer than wild-type RPS7A, were approximately 70-80% lower than those of wild-type RPS7A across all backgrounds (Fig 2C).

2) On page 13, line 199, the Rps7A protein level in wt is not shown. This is because wt is not HA-tagged, but it would be interesting to have a relative quantification of Rps7A levels in Rps7A-CgHIS3 with respect to those in wt.

We agree with your idea. Unfortunately, we don’t have an antibody against Rps7A, which makes experimentation difficult. Based on the mRNA levels, Rps7A from RPS7A::CgHIS3 may be reduced compared to that from wild-type RPS7A.

3) Page 15, line 228. There are two ... at the end of the line.

Thank you for pointing out. We deleted one “.” Please refer to page 16, line 280.

4) Page 17, line 268. “we deleted this sequence...”. It is important to know which sequence was deleted. Since there is a putative 5 bp overlap between the Rap1 and Fhl1 binding sites, if the entire sequence is deleted the consequence would be a lack of both binding sites. If only part of it is removed, Rap1 (or Fhl1) may still bind to it. According to line 272 on page 18 both sites were removed but this conflicts with other conclusions given later. See point #5 below.

We deleted the entire sequence covering the Rap1p and Fhl1p binding sites, thus both binding sites. We revised the sentence to clarify this point, and add a graphical representation of mutated sequences in Fig 5A.

Page 18, lines 339-340:

We deleted this entire sequence covering the Rap1p and Fhl1p binding sites to make a mutated version of the RPS7A promoter (Fig 5A, rps7a-mut).

Pages 41–42, lines 860-869

The right panel shows enlarged sequences of the dotted line areas from the left panel. The numbers represent the upstream region relative to the transcription start site. The simultaneous binding sites for Rap1p (RPS7A: 5'-ATGTTTGGGT-3'; RPS7B: 5'-AGGTACGGAT-3') and Fhl1p (RPS7A: 5'-ACCCAAACAT-3'; RPS7B: 5'-ATCCGTACCT-3') are highlighted in red, while the Fhl1p binding site alone in RPS7B (5'-ATCCGTACAT-3') is shown in orange.

5) Line 272 states that (Fig. 5B) the lack of both Rap1/Fhl1 binding sites is not affecting Rps7A expression (as demonstrated by the GFP reporter). This is quite surprising because these factors are described as very important for RP gene expression in many previous papers. Here it appears that in the RPS7A gene they are only required for negative feedback of the Rps7 protein on RPS7A transcription. Although the question is discussed below (pages 23-24) it is quite surprising and as no potential mechanism is suggested it leaves the question too unresolved. Is the Rps7 protein interacting with Rap1 (or Fhl1, or another factor) to prevent binding? If so, the experiment shown below (fhl1 mutant) will demonstrate that Fhl1 is NOT the target of Rps7A. As for the function of Rap1, since Rap1 is essential a ts mutant or Auxin-degron can be used to test the function of Rap1 in Rps7A transcription. Alternatively the rap1delta-sil mutant has been used in the context of RP gene expression where it affects the RNA pol II transition between initiation and elongation (doi:10.1371/journal.pgen.1000614).

We appreciated your variable comment and discussion. We aim to express ideas with confidence only when supported by robust evidence. Currently, we unfortunately lack comprehensive data on the regulatory mechanism governing the RPS7A gene, which is primary involved in the negative feedback of Rps7 protein on RPS7A transcription. Nevertheless, as discussed in pages 25-26, lines 509–524, we need to suppose Fhl1p's function independent of the Fhl1p site in the RPS7A promoter region to explain the effect of the rps7a-mut fhl1∆ double mutant, and this is the utmost mechanistic suggestion we can make at this stage. While we refrain from further speculating on potential interactions with other regulators in the present manuscript, we included the following description in discussion.

Page 27, lines 544-549:

Moreover, Rps7Ap may directly interact with a transcriptional regulator to achieve RPS7A-specific feedback regulation, as known by the general autoregulatory mechanism of RP production [3]. At least, Fhl1p is not the target of Rps7A (Fig 5), indicating that Rap1p is the prime target identified so far. While Rap1p is essential, the use of a ts-mutant [47,48] or Auxin-degron system, and the rap1∆sil mutant [49] offers valuable tool for further investigation.

6) page 24 line 373. RPS7 is RPS7B?

We intended for “RPS7” to refer to both RPS7A and RPS7B. To clarify, we revised the term to “RPS7A/B”. Please refer to page 26, line 516.

7) In Discussion (page 24) the authors describe the existence of other TFs that can be activating Rps7A promoter, such as Ifh1 or Hmo1. I think that Sfp1 should be included because it another well-known transactivator of many RP genes.

Thank you for your suggestion. We revised the manuscript to include Sfp1 as one of the other potential TFs.

Page 26, lines 517-521:

This suggests that the Fhl1p site in the RPS7A promoter is dispensable for Fhl1p function, as Fhl1p can still be recruited to the RPS7A promoter via interactions with other transcription factors, such as Rap1p (a pioneer transcription factor [42]) and Sfp1p (a transcription factor requited by Rap1p [40]), and/or Hmo1p (an HMG family chromatin interactor [43]), which associated with the RPS7A promoter region by themselves [40,43,44].

Reviewer #2: The authors focused on a very interesting topic: what does it happen with paralogous genes? how redundant/specific are they? They have focused on a pair of ribosomal proteins (RPS7A and RPS7B). The main problem of the paper is that the main aim of the study, the specific objectives and the hypotheses are never clearly stated upfront. Therefore, the reader does not know if the described experiments make sense in order to achieve that aim and answer those research questions.

Thank you for your valuable feedback. Based on your comments, we revised the entire manuscript, with particular focus on the results section, to provide the objectives and/or hypotheses presented more clearly at the outset. Please refer the corresponding parts as follows.

Page 13, lines 219-223:

Paralogous RP gene expression is regulated by multiple layers of transcriptional and post-transcriptional levels [3]. In trypanosomatids, recent findings show that the translation of RP genes can be regulated by RNA-binding proteins interacting with their 5'- or 3'-UTRs [27], contributing to paralog-specific expression [28]. Similarly, in yeast, UTRs may serve as regulatory elements for RP paralog expression, potentially playing a more significant role than previously anticipated.

Page 14, lines 242-243:

Collectively, the 3'-UTR of RPS7A seems to have some cis element(s) to enhance the basal transcription and/or stability of RPS7A mRNA.

Page 14, lines 248-251:

The presence or absence of an intron in a tRNA gene influences its ability to function as an insulator, preventing the spread of silenced chromatin and thereby altering the silencing status of nearby pol II genes [29–31]. Thus, it is possible that the tRNA intron in tL(CAA)N additionally affect RPS7B expression in the tl(caa)∆int* strain. To verify whether…

Page 16, lines 280-281:

And, the tRNA intron had no effect on RPS7B expression with the long 3'-UTR.

Page 16, lines 294-296

According to Ghulam et al. [7], Rps7p expression is regulated by both transcriptional and translational levels. To assess the transcriptional effect of RPS7B, we first performed semi-quantitative RT-PCR.

Page 17, lines 306-308:

We then examined whether the tl(caa)Δint* strain impaired RPS7B mRNA expression post-transcriptionally, hypothesizing that mRNA abundance directly reflects protein levels due to aberrant 3'-UTR.

Page 18, lines 327-334:

In the traditional view, regulation of RPs often occurs at protein level or in response to the amount of assembled ribosomes, with feedback process frequently leading to transcriptional repression of RP genes [32]. Thus, RPS7 paralog-specific feedback response would initiate at transcriptional level, and in this process, the upstream and/or downstream regions of the ORF may provide clues to understanding the RPS7-specific gene regulation mechanism in vivo. To explore difference of transcriptional features between RPS7A and RPS7B genes, we first focused on the upstream region and searched for known cis elements located in the promoter region of RPS7A and RPS7B on Saccharomyces Genome Database.

Page 20, lines 380-385

Collectively, the upstream region, particularly the Fhl1p and/or Rap1p binding sites, play a crucial role in RPS7 paralog specific response.

Secondly, to evaluate the impact of the downstream region on their paralog-specific expressions, we additionally generated reporter genes in which constructs share a common ADH1 promoter and GFP-FLAG-HIS3 gene, with the only difference being the 3'-UTRs and terminators (RPS7A or RPS7B), in contrast to the upstream region.

Page 20, lines 390-392

Since Fhl1p and/or Rap1p binding sites are crucial for RPS7-paralog specific response (Fig 5B), we generated an fhl1∆ strain and repeated the reporter assay to analyze whether Fhl1p acts on RPS7 paralog expression.

Page 21, lines 414-416

Thus, the different contributions of Fhl1p (and/or Rap1p) to the RPS7-paralog promoter regions are key determinants of RPS7-paralog expression, which is influenced by the levels of Rps7Ap (or total Rps7p).

In my opinion, some of the presented experiments seem irrelevant in order to understand the relationship between RPS7A and RPS7B. For example, the authors constructed a strain with an artificially long 3'UTR in RPS7B. Then, they demonstrate that NMD controls the expression of that modified RPS7B gene. However, I wonder why this experiment is relevant if this never occurs in natural/biotechnological strains. WT RPS7B has a normal 3'UTR which is probably not targeted by NMD. Therefore, the implications of these results are totally obscure to me.

Thank you for your opinion. We originally interested in tRNA biology and began the manuscript with our unexpected (artificial) mutant and the observation made from this mutant. While the biological relevance of our mutant is limited, considering that tRNA gene regions are often targeted by transposon (Guimarães, et al., 2021, Front. Microbiol., 12, 634004, doi: 10.3389/fmicb.2021.634004), such genomic changes altering the 3'-UTR of RPS7B could occur during evolution (and were probably eliminated by natural selection). Therefore, we believe our serendipitous finding holds significance, at some level, within the context of the manuscript. At this point, we added this to the discussion. Please refer to the following.

Page 22, lines 426-431:

Our serendipitous finding began with an unexpected genomic change near a tRNA gene during construction of the tl(caa)∆int strain. Although our case is completely artificial, insertion are often introduced at genomic fragile sites by transposons [34]. Indeed, tRNA loci exhibit mutation rates approximately 7 to 10 times greater than the genome-wide average [35]. Thus, genomic changes or mutations similar to our mutant might arise under natural conditions during evolution.

Moreover, other comparisons do not seem to be absolutely fair. In my opinion, it is very interesting to know how the amount of mRNA and transcript of one of the paralogs is affected when the other paralog is knocked down or overexpressed. Here, the authors have looked at the effect of the RPS7A deletion on RPS7B. They have also looked at the effect of the RPS7B deletion on RPS7A. However, most of the quantitative analyses on RPS7A where done with the long-UTR RPS7B gene. In this experiment, the transcription of RPS7B is not prevented, but the mRNA is mostly degraded and hence never translated. The reciprocal experiment (RPS7A targeted by NMD) is never done. In order to rule out symmetrical effects, the analyses need to comprehensive.

We understand your points, and we agree that further comprehensive analyses are needed to rule out these symmetrical effects. As mentioned previously, the story presented in this manuscript began with our spontaneous mutant, which has RPS7B gene with the long 3'-UTR. This led us to focus more specifically on the location where the RPS7B gene derivatives are expressed. Further, according to the genomic context, there are no tRNA genes or fragile sites around RPS7A, suggesting that natural insertion would not occur. Therefore, we did not investigate the effect of a 3'-UTR insertion in the RPS7A 3'-UTR. We have added these descriptions including the limitations of our experiment.

Pages 24-25, lines 483-493:

We did not investigate the effect of a 3'-UTR insertion in the RPS7A 3'-UTR, specifically a 2.3-kb insertion of loxP::CmR::hisG::loxP, as seen in rps7b-102. Given the genomic context, there are no tRNA genes or fragile sites around RPS7A, suggesting that natural insertion would not occur. However, more comprehensive studies are needed to examine the impact of 3'-UTR insertions in RPS7A to better understand the overall imbalance in regulation between paralogous genes.

Finally, I am not entirely satisfied by the way that the statistical analyses are performed, reported and interpreted.

1) I did not see information on the number of replicates used in each experiment; n must be reported.

Thank you for pointing out. We described the number of replicates (n) for each experiment in the main text or in the figure legends.

2) t-test is used in most analyses, but this is incorrect because many of those involve multiple comparisons. If all comparisons are of interest, the authors should use ANOVA plus a post hoc test (e.g. Tukey's test). If only comparisons to a single control are of interest, Dunnet's test should be used instead. If not all the comparisons are of interest and there is no common control, the authors should correct the p-values of the t-test for multip

---

## [Decision Letter · Decision Letter 1]

24 Apr 2025

PONE-D-25-01723R1Beyond the ORF: Paralog-specific regulation of RPS7/eS7 mRNAs via 3'-UTRs and promoter sequencesPLOS ONE

Dear Dr. Hayashi,

Thank you for submitting your manuscript to PLOS ONE. After careful consideration, we feel that it has merit but does not fully meet PLOS ONE’s publication criteria as it currently stands. Therefore, we invite you to submit a revised version of the manuscript that addresses the points raised during the review process.

**Reviewer 3 has raised a specific concern that must be addressed before the manuscript can be accepted for publication.**

**I kindly request you to respond to the comment(s) and resubmit the revised manuscript.**

**I look forward to receiving your revised submission.**

We look forward to receiving your revised manuscript.

Kind regards,

Koppolu Raja Rajesh Kumar, PhD

Academic Editor

PLOS ONE

Journal Requirements:

Reviewers' comments:

Reviewer's Responses to Questions

**Comments to the Author**

1. If the authors have adequately addressed your comments raised in a previous round of review and you feel that this manuscript is now acceptable for publication, you may indicate that here to bypass the “Comments to the Author” section, enter your conflict of interest statement in the “Confidential to Editor” section, and submit your "Accept" recommendation.

Reviewer #1: All comments have been addressed

Reviewer #3: (No Response)

2. Is the manuscript technically sound, and do the data support the conclusions?

Reviewer #1: Yes

Reviewer #3: Yes

3. Has the statistical analysis been performed appropriately and rigorously? 

Reviewer #1: I Don't Know

Reviewer #3: Yes

4. Have the authors made all data underlying the findings in their manuscript fully available?

Reviewer #1: Yes

Reviewer #3: Yes

5. Is the manuscript presented in an intelligible fashion and written in standard English?

Reviewer #1: Yes

Reviewer #3: Yes

6. Review Comments to the Author

Reviewer #1: All questions have been solved or argued. The paper can be published in the current version in PLOS One

Reviewer #3: The authors' revision is much improved. I only have one request. As I mentioned in the first review, it is technically possible that the increase in RPS7B RNA levels that occurs upon deletion of RPS7A could reflect duplication of the chromosome on which RPS7B is found (aneuploidy). The authors should explicitly acknowledge that in the discussion. Currently, they only vaguely hint at the possibility.

7. PLOS authors have the option to publish the peer review history of their article (what does this mean? ). If published, this will include your full peer review and any attached files.

**Do you want your identity to be public for this peer review?** For information about this choice, including consent withdrawal, please see our Privacy Policy .

Reviewer #1: No

Reviewer #3: No

---

## [Author Response · Author response to Decision Letter 2]

25 Apr 2025

Point-to-point answers

5. Review Comments to the Author

Reviewer #3: The authors' revision is much improved. I only have one request. As I mentioned in the first review, it is technically possible that the increase in RPS7B RNA levels that occurs upon deletion of RPS7A could reflect duplication of the chromosome on which RPS7B is found (aneuploidy). The authors should explicitly acknowledge that in the discussion. Currently, they only vaguely hint at the possibility.

Thank you for your comment. We revised the relevant section of the manuscript as follows.

Page 23, line 366:

…deletion of one of RP paralog pairs can lead pivotal effects on growth

Page 23, lines 368-370:

Therefore, in the case of Rps7p regulation, the increase in RPS7B mRNA levels that observed in rps7A∆ is possibly due to aneuploidy, specifically a duplication of the chromosome that carries RPS7B.

---

## [Editor Report · Decision Letter 2]

27 Apr 2025

Beyond the ORF: Paralog-specific regulation of RPS7/eS7 mRNAs via 3'-UTRs and promoter sequences

PONE-D-25-01723R2

Dear Dr. Hayashi,

We’re pleased to inform you that your manuscript has been judged scientifically suitable for publication and will be formally accepted for publication once it meets all outstanding technical requirements.

Kind regards,

Koppolu Raja Rajesh Kumar, PhD

Academic Editor

PLOS ONE
---

## [Editor Report · Acceptance letter]

PONE-D-25-01723R2

PLOS ONE

Dear Dr. Hayashi,

I'm pleased to inform you that your manuscript has been deemed suitable for publication in PLOS ONE. Congratulations! Your manuscript is now being handed over to our production team.

Kind regards,

on behalf of

Dr. Koppolu Raja Rajesh Kumar

Academic Editor

PLOS ONE